# A Novel Preparation of Ag Agglomerates Paste with Unique Sintering Behavior at Low Temperature

**DOI:** 10.3390/mi12050521

**Published:** 2021-05-06

**Authors:** Junlong Li, Yang Xu, Ying Meng, Zhen Yin, Xuelong Zhao, Yinghui Wang, Tadatomo Suga

**Affiliations:** 1Institute of Microelectronics of Chinese Academy of Sciences, Beijing 100029, China; lijunlong@ime.ac.cn (J.L.); mengying@ime.ac.cn (Y.M.); yinzhen@ime.ac.cn (Z.Y.); zhaoxuelong@ime.ac.cn (X.Z.); wangyinghui@ime.ac.cn (Y.W.); 2University of Chinese Academy of Sciences, Beijing 100049, China; 3Institute of Microelectronic Technology of Kunshan, Suzhou 215347, China; 4Collaborative Research Center, Meisei University, Hino-shi, Tokyo 191-8506, Japan; suga@gakushikai.jp

**Keywords:** Ag agglomerates, sintering, Pt-catalyzed formic acid, low temperature

## Abstract

A novel bonding process using Ag agglomerates paste prepared by Ag_2_O reduction has been proposed, which solved the problem of Cu substrate oxidation in the conventional Ag_2_O sintering process for Cu–Cu bonding. By applying the Ag agglomerate paste to Ag–Ag bonding, a shear strength of 28.3 MPa at 150 °C was obtained. Further studies showed that the optimum sintering temperature was at 225 °C, and a shear strength of 46.4 MPa was obtained. In addition, a shear strength of 20 MPa was obtained at 225 °C for Cu–Cu bonding. Compared to common Ag pastes, the results in this paper revealed that the sintering behavior of Ag agglomerates was unique, and the sintering mechanisms for Ag–Ag and Cu–Cu bonding were also discussed.

## 1. Introduction

Third-generation semiconductors, such as gallium nitride and silicon carbide, are characterized by high pressure, high frequency and high temperature compared to silicon, and have been widely used in the fields of rail transit, industrial control, automotive electronics and photovoltaic power generation [1,2]. With the increasing in power density of devices, the requirement of reliability in packaging technology becomes harsher. Conventional high-temperature solders, such as Zn-based [3,4] and Au-based [4,5], are difficult to meet the demand for the reliability of power devices under high-temperature operating conditions due to their own drawbacks. Thus, it is necessary to develop a new packaging technology to meet the requirements.

A large number of studies for the use of metallic nanoparticles as bonding materials, such as silver and copper, have demonstrated their great potential in high-temperature packaging technologies in the future [6,7,8,9]. It is known that the apparent melting point of metallic particles becomes much lower than that of their bulk material as the size decreases to several nanometers [10]. While the melting points of sintering layers will approach that of bulk materials once the connection between the nanoparticles has formed [11,12,13]. Furthermore, silver has a higher thermal and electric conductivity than the conventional solders. Therefore, the bonding processes by silver-organic nanoparticles sintering have been developed. However, the organic layers coating the Ag nanoparticles seriously hindered the connection between the nanoparticles [7,12,14,15]. Therefore, it is not conducive to low-temperature bonding.

Ag_2_O microparticles, which have a much lower cost than Ag nanoparticles, have been proposed in the metal–metal bonding process [16,17]. Ag nanoparticles can be formed from Ag_2_O microparticles through the redox reaction with solvents, such as DEG (diethylene glycol), TEG (triethylene glycol) and PEG (polyethylene glycol) [18,19,20,21]. In particular, DEG and TEG could cause the copper substrates to be oxidized while PEG could suppress the oxidation of the copper substrates but cause the void existing in the sintering layer [20]. In addition, PEG could reduce the natural oxide film on the copper substrates, but the sintering temperature was 300 °C due to the high boiling point of the solvent [22].

In our previous study, Ag_2_O paste has been used for the Ag–Cu bonding at low temperatures [23]. By reducing the copper oxide layer, Pt-catalyzed formic acid vapor can improve the adhesive properties between the sintered layer and the Cu substrates by highly reactive hydrogen radicals. However, there is still a risk for the copper-plated SiC chips bonding process due to the formation of copper oxide during the sintering process. Therefore, it is momentous to solve the problem for the application of Ag_2_O paste.

In this paper, Ag agglomerates paste was prepared by reducing Ag_2_O powder in Pt-catalyzed formic acid vapor. The features of the morphology and components of Ag_2_O powder before and after the treatment were investigated. Due to the unique structure, the sintering behavior and properties of Ag agglomerates at different temperatures were investigated. Moreover, the shear strength of Cu–Cu bonding was significantly enhanced by this method and the mechanism was also discussed.

## 2. Experimental

### 2.1. Materials

The Ag_2_O powder was purchased from Shanghai Buwei Applied Materials Technology Co., Ltd. (Shanghai, China). The ethylene glycol (EG, 99%) and anhydrous formic acid (99%) were purchased from Shanghai Macklin Biochemical Co., Ltd. (Shanghai, China). The flaked Ag agglomerates were prepared by Ag_2_O microparticles treated in the Pt-catalyzed formic acid vapor at 50 °C for 5 min. Then a special amount of EG mixed with flaked Ag agglomerates powder. For Cu–Cu bonding experiments, SiC chips (6 mm × 6 mm × 0.35 mm) were chosen as the dummy chips and were sputtered with Ti (20 nm) and Cu (300 nm) successively. Oxygen-free Cu cylindrical discs (a diameter of 10 mm and a height of 3 mm) were used as the lower Cu substrates, and for the Ag–Ag bonding experiments, the Ag-plated Cu discs were used. The smaller Cu discs with a diameter of 6 mm and a height of 3 mm were used as upper substrates, and the larger Cu discs with a diameter of 10 mm and a height of 3 mm were used as lower substrates. Before the sintering process, the chips and discs were ultrasonically cleaned in acetone, ethanol and citric acid.

### 2.2. Methods and Characterizations

The prepared Ag paste was printed on the lower substrates using a steel mask with a thickness of 100 μm, followed by mounting the SiC chips or upper substrates. Firstly, the assembled sample was preheated at 150 °C for 5 min, and then sintered at different temperatures for 30 min under 5 MPa with a heating rate of 2 °C/s in a flowing N_2_ gas with the Pt-catalyzed formic acid vapor, as illustrated in Figure 1.

The shear strengths were measured by a shear strength tester (TRY Precision MFM 1200) with a speed of 50 μm/s. For every condition, 5 specimens were measured. Thermogravimetric–differential scanning calorimetry (TG–DSC) was employed to investigate the thermal characteristics of the Ag paste in flowing N_2_ gas at a heating rate of 10 °C/min. Scanning electron microscopy (SEM, JEOL JSM-7800F), X-ray diffraction (XRD, D8-Advance, Bruker, Germany) and X-ray photoelectron spectroscopy (XPS, Thermo escalab 250XI, Thermofisher, Waltham, MA, USA), equipped with a monochromatic Al X-ray radiation source (1486.6 eV), were used to verify the difference in Ag_2_O microparticles before and after the treatment in Pt-catalyzed formic acid vapor, respectively. The adsorbed adventitious carbon (C1s, 284.8 eV) was used as a reference for the calibration of binding energy.

## 3. Results and Discussion

### 3.1. Characteristics of Ag Paste

Figure 2 shows the morphologies of the Ag_2_O powder before and after the treatment in Pt-catalyzed formic acid vapor. Most of the Ag_2_O microparticles were irregular and spherical, and the size of the microparticle was about 400–1500 nm. It is worth mentioning that the surface of the Ag_2_O microparticle was completely covered with a layer of Ag nanoparticles, as shown in Figure 2b, and the average diameter of the nanoparticles was around 10 nm. It is inferred that the Ag nanoparticles were formed by the decomposition of Ag_2_O microparticles during storage. The existence of Ag nanoparticles was further confirmed by XPS analysis. Although the surface energy of the tiny-sized Ag nanoparticles is theoretically extremely high, the diffusion rate between them was slow at a low temperature, resulting in the formation of a small amount of larger Ag nanoparticles. In contrast, the Ag_2_O microparticles transformed into flaked agglomerates with loose and porous structures consisting of Ag nanoparticles after the treatment in Pt-catalyzed formic acid vapor, as shown in Figure 2c,d. In order to further verify the compositional and structural changes of the Ag_2_O powder, XRD and XPS analysis were conducted.

As shown in Figure 3, the crystal phase of the Ag_2_O powder before and after the treatment was investigated by XRD. The diffraction pattern for the Ag_2_O powder exhibits four distinct peaks at 32.8°, 38.0°, 54.9° and 65.5° in Figure 3a, corresponding to (111), (200), (220) and (311), respectively. However, the diffraction pattern of the Ag_2_O powder after treatment shows five peaks at 38.1°, 44.3°, 64.4°, 77.5° and 81.5°, which are the characteristics of Ag, as shown in Figure 3b. It indicates that the Ag_2_O microparticles were reduced into Ag agglomerates after the treatment in Pt-catalyzed formic acid vapor. In more detail, no characteristic diffraction peaks belonging to Ag were detected in Figure 3a. On one hand, the crystallization of the Ag nanoparticles was weak, resulting in the poor diffusion capacity of the Ag atoms in Figure 2b. On the other hand, it is implied that the Ag nanoparticles were well dispersed on the surface but relatively low in content. Therefore, it is inferred that the Pt-catalyzed formic acid vapor could effectively reduce the Ag_2_O powder in a shorter time and at a lower temperature. In addition, the crystallinity and diffusion capacity of the Ag nanoparticles was improved in Figure 2d.

Further characterization by XPS shows that there were significant changes in the elemental Ag and O valence states of the Ag_2_O powder before and after treatment in the Pt-catalyzed formic acid vapor. Figure 4a clearly illustrates that the binding energies of the Ag3d_5/2_ and Ag3d_3/2_ electrons in the Ag_2_O powder were 368.4 eV and 374.4 eV, respectively. Obviously, the existence of Ag nanoparticles was evidenced by two peaks at 368.2 eV and 374.2 eV, which correspond to binding energies of the Ag3d_5/2_ and Ag3d_3/2_ electrons of Ag, respectively [24]. Therefore, the elemental Ag mainly existed in Ag^0^ and Ag^-1^ valence states for the Ag_2_O powder. The O1s spectra in Figure 4b show two peaks at 529.2 eV and 531 eV, corresponding to O^−2^1s and O^0^1s, respectively. It indicates the absence of H_2_O adsorption on the Ag nanoparticle surface. However, the intensity of the peaks of the Ag_2_O powder at Ag3d and O^−2^1s were much weaker, which indicates that the surface of the Ag_2_O microparticle was compactly covered by Ag nanoparticles. However, the Ag3d peaks belonging to Ag_2_O disappeared after the treatment with Pt-catalyzed formic acid vapor, and elemental Ag was present only in the Ag^0^ valence state, as shown in Figure 4c. Meanwhile, there were no O^−2^1s peaks belonging to Ag_2_O in the O1s spectra, as shown in Figure 4d. These results indicate that the Ag_2_O had been reduced efficiently by H radicals. It is notable that the peak at the binding energy equaling to 533.2 eV was significantly enhanced. This indicates that there was a large amount of H_2_O formed during the reduction process. In addition, it can also be inferred that the hydrophobic property of the Ag nanoparticles led to their rearrangement to form flaked agglomerates with a loose structure.

By mixing the Ag agglomerates powder with EG (70 wt%), the Ag paste was prepared. Prior to the sintering process, the thermal property of the Ag paste was investigated by TG–DSC analysis. Figure 5 shows the TG–DSC curves of the Ag paste heated in N_2_. It can be seen that both the TG and DSC curves of the Ag paste decreased significantly at 150 °C. Except for an endothermal peak in the DSC curve, there were no other peaks during the continuous increase in the temperature. It means that there was no redox reaction between the residual Ag_2_O and the EG solvent. Correspondingly, the TG curve shows that the weight loss in the range from 100 °C to 150 °C was related to the fast evaporation of the EG solvent.

### 3.2. Sintering Processes and Mechanism for Ag–Ag Bonding

Firstly, Ag–Ag bonding experiments were performed on the substrates with a Ag layer to investigate the sintering behavior of Ag paste. As shown in Figure 6a, the average shear strength of Ag–Ag bonding specimens was 28.3 MPa at the sintering temperature of 150 °C. When the sintering temperature was 225 °C, the shear strength was increased to the maximum value of 46.4 MPa. However, the shear strength decreased gradually with the further increase in sintering temperature, which indicates that the sintering structure changed. In addition, the influence of different preheating temperatures on the shear strength were also investigated, as shown in Figure 6b. When the sintering temperature was 200 °C, it was found that the difference in temperature of the preheating stage did not have a significant effect on the shear strength. This indicates that the sintering behavior of Ag agglomerates during the preheating stage was preferred to the diffusion between agglomerates rather than that between the sintered layer and the substrate, due to the hindering effect of the EG solvent.

In order to verify the sintering properties of Ag paste, the fracture surfaces of the specimens sintered at different temperatures were observed by SEM, as shown in Figure 7. After sintering at 150 °C and 175 °C, ductile fracture traces in the fracture surfaces were not very obvious, and there were a lot of large grains as shown in Figure 7a,b. It indicates that tiny-sized Ag nanoparticles tended to be sintered together to form large grains. After sintering at 200 °C and 225 °C, the size of the grains shrank and the morphologies of the grains became uniform with the increase in the sintering temperature, as shown in Figure 7c,d, respectively. In addition, more ductile fracture traces appeared in the fracture surfaces, and the densification of the sintered layers was notably improved. However, the cracks appeared on the interior of the sintered layer when the sintering temperatures were 250 °C and 275 °C, as shown in Figure 7e,f, respectively. Therefore, the sintering behavior of Ag agglomerates at relatively higher sintering temperatures significantly influenced the sintering properties.

Figure 8 shows SEM images of the cross-sectional structure of specimens sintered at different temperatures. Obviously, there were significant differences in the variation in pore size in the sintered layers. Overall, the densification of the sintered layers decreased and then increased with the increase in the sintering temperature, which would be discussed in the sintering mechanism. Correspondingly, the pore distribution reached the maximum at 175 °C and significantly reduced with further increase in the sintering temperature. More importantly, it was found that the size of the grains and the width of the grain boundaries gradually increased at 225 °C and 275 °C. This is due to the different atomic diffusion patterns at different temperatures. At lower temperatures, the atomic diffusion was dominated by surface diffusion and grain boundary diffusion, resulting in the formation of inhomogeneous grains. At higher temperatures, grain boundary migration was activated, which led to a gradual increase in grain size.

It is known that dynamic grain growth usually takes place during the preheating stage and within the first few minutes after reaching the sintering temperature. In order to investigate the dynamic growth of Ag agglomerates, the morphologies of the Ag paste sintered at different temperatures for 5 min were observed by SEM, as shown in Figure 9. As mentioned above, porous structures consisting of Ag nanoparticles were formed after the treatment in Pt-catalyzed formic acid vapor for 5 min. At 100 °C, the agglomerates transformed into large-sized Ag nanoparticles with twin planes in order to eliminate the intra-agglomerate pores [25], and these Ag nanoparticles were interconnected to form hollow sintered structures. At 150 °C, the diffusion of atoms on the surface of hollow sintered structures was accelerated, resulting in the decrease in size of the pores in the surface of the hollow sintered structure. However, at 175 °C and 225 °C, the sintered structures transformed into porous and loose structures again. This is due to the fact that as the sintering process proceeded, the hollow sintered structure shrunk to the inside, and the pores inside the structure had to be diffused outside to achieve the densification process of the sintered structure. At 200 °C, 250 °C and 275 °C, the pores on the surface of the sintered structures gradually were eliminated again.

Based on the results, we analyzed the sintering mechanism of Ag agglomerates as shown in Figure 10. When Ag_2_O microparticles were treated in Pt-catalyzed formic acid vapor, the Ag nanoparticles inside the agglomerates developed a directional distribution due to the hydrophobic properties, as shown in Figure 10a. Due to the extremely high surface energy, these tiny-sized Ag nanoparticles would be sintered together easily, as shown in Figure 10b. As a result, a large number of twin planes appeared in the crystal structure of the Ag nanoparticle formed by sintering, as shown in Figure 9c. Moreover, the hollow sintered structure means that the concentration of vacancies inside the sintered structure was very high. As the sintering temperature increased, the Ag atoms gradually eliminated the pores on the surface of the hollow sintered structure by surface diffusion and grain boundary diffusion, as shown in Figure 10c,d. Eventually, only a small number of pores remained at the position of multiple grain junctions, as shown in Figure 9d. However, the pores inside the sintered structure still existed, which limited the densification process. The internal pores diffused to the outside of the sintered structure by increasing the concentration of vacancies in the grain boundaries, leading to the formation of irregular pores at the positions of grain boundaries, as shown in Figure 9e. Subsequently, the re-formed pores were gradually eliminated by surface diffusion and grain boundary diffusion at 200 °C, as shown in Figure 9f. Moreover, a similar process occurred again during the sintering temperature increase from 225 °C to 275 °C, as shown in Figure 9g–i.

Since surface diffusion and grain boundary diffusion were performed at the expense of the total atomic number of grains, it led to a gradual decrease in grain size during the densification process of the sintered structures. As a result, smaller sized grains gradually formed pores as the atomic number decreased, which assisted the movement of the internal pores to the outside, as shown in Figure 10e. Due to the pegging effect of the pores, grain boundary migration cannot be activated at lower temperatures. When the pores were gradually eliminated, grain boundary migration played a dominant role. By means of grain boundary migration, the atoms in the smaller grains transferred into the larger grains, resulting in a rapid increase in grain size for the large grains. However, due to the presence of more vacancies in the grain boundaries during migration, the strength of the connection at the grain boundary positions was reduced [10]. Therefore, most of the cracks appeared at the positions of grain boundaries, as shown in Figure 8f and Figure 10f.

## 4. Sintering Processes and Mechanism for Cu–Cu Bonding

As shown in Figure 11a, the copper layer of the SiC chips was oxidized and peeled off during the shear test for Cu–Cu bonding using Ag_2_O paste. This is caused by the reaction between Ag_2_O and the EG solvent, and the formation of a large amount of H_2_O during the sintering process as given in Equation (1), resulting in oxidation of the copper layer. Since the thickness of the copper layer is 300 nm, the Ag_2_O paste exhibited a significant risk to the Cu–Cu bonding process. For the Ag paste, this problem could be solved easily and the chemical reaction is given in Equation (2). After the shear test, it was obviously seen that the copper layer was not oxidized and peeled off.
(1)$$7{\mathrm{Ag}}_{2}O+C_{2}H_{6}O_{2}\to {14\mathrm{Ag}+2\mathrm{CO}}_{2}↑{+3H}_{2}O↑$$
(2)$${\mathrm{Ag}}_{2}O+2H\;(\mathrm{radical})\to {2\mathrm{Ag}+H}_{2}O↑.$$

Figure 12 shows the average shear strength with the standard deviation of the sintered specimens of the copper-plated SiC chips and the bare copper substrates. It indicates that the shear strength of the specimens increased with the upregulation of thesintering temperature. At 200 °C, the shear strength was 10 MPa. While the temperature was upregulated to 225 °C, the shear strength increased to 21 MPa, indicating that the mechanical strength can meet the requirement of practical application. At a sintering temperature of 275 °C, the sintering strength can reach 31 MPa. Based on the experimental results of Ag–Ag bonding, the Ag sintered structures formed at 225 °C were more conductive for the enhancement of shear strength. Therefore, the improvement in the shear strength effect was more significant at 225 °C. Since the experimental results of Cu–Cu bonding were obviously different from those of Ag–Ag bonding, and the overall shear strength was weak, the reason is mainly due to the fact that the connection between the sintered layer and the surface of the SiC chip or bare Cu substrate was affected by the surface structure. At lower temperatures, the formation of a metallic bond between the sintered layer and the substrate was less effective. Therefore, it is necessary to conduct an in-depth analysis of its sintering mechanism.

In order to explain this, the fracture surfaces were investigated by SEM as shown in Figure 13. At 200 °C, it is clearly seen that the Ag agglomerates contacted with the upper substrate, despite the diffusion of Ag atoms on the surface of the copper layer, was not significant as shown in Figure 13a. The rate of diffusion increased obviously with the increase in sintering temperature. In addition, the size of the pores on the fracture surface were also significantly reduced, as shown in Figure 13b,c. Importantly, there were a large amount of ductile fracture traces on the fracture surface at a sintering temperature of 275 °C. Meanwhile, it was also observed that the grain size increased and became uniform owing to the diffusion of the grain boundary in Figure 13d. According to the fracture position in the sintered layer after the shear test, it is thought that the differences in shear strength were caused by different interfacial adhesive strengths between the sintered layer and the upper substrate.

Based on the above results, the sintering mechanism of Ag agglomerate paste for Cu–Cu bonding was analyzed, as shown in Figure 14a. Since two types of Cu surface were involved in the bonding experiments and there was a significant difference in the strength of the adhesion between the two type of substrates and the sintered layer, it is necessary to discuss the difference between the two types of copper layer structures. There are two main diffusion patterns for the atoms on the surface of the Ag nanoparticle at the interface between the sintered layer and the substrate [26]. One pattern is to diffuse to the surface of adjacent Ag nanoparticles, which forms a neck between the Ag nanoparticles. It leads to the enhancement of the strength of the sintered structure and promotes the densification of the sintered structure. The other pattern is the surface diffusion on the substrate to expand the contact area.

The atoms on the surface of the Ag nanoparticles diffused rapidly at the interface, forming a connection region. However, there were a large number of grain boundaries in the surface of the bare copper substrates, so the Ag atoms not only formed connections with the Cu atoms in the contact area, but also diffused through the vacancies in the grain boundaries to the inner part of the lower substrate, as shown in Figure 14b. Therefore, it results that the adhesive strength was much higher than that between the upper substrate and the sintered layer [26,27]. For the copper layer of the upper substrate, there are almost no larger grains and a large number of grain boundaries on the surface structure. So, the Ag atoms could not diffusion through the grain boundaries to the interior of the copper layer of the upper substrate. This results in a weak strength between the sintered layer and the upper substrate at lower temperatures, and the fracture occurred just at the interface between the sintered layer and the upper substrate. However, since grain boundary migration activated in the sintered layer at 275 °C, it enhanced the diffusion rate of Cu atoms to the grain boundaries in the sintered layer. Therefore, the diffusion of Cu atoms to the grain boundaries in the sintered layer was accelerated and a large number of Cu atoms filled the vacancies in the grain boundaries, resulting in the formation of a denser sintered structure at the interface. Meanwhile, a number of sintered structures remained on the surface of the upper substrate. In addition, the position of the fracture surface moved towards the interior of the sintered layer.

## 5. Conclusion

In summary, the problem of Cu substrate oxidization during the sintering process of conventional Ag_2_O pastes has been solved by reducing Ag_2_O microparticles to Ag agglomerates in Pt-catalyzed formic acid vapor. For Cu–Cu bonding, a shear strength of 21 MPa was obtained at 225 °C for 30 min under a pressure of 5 MPa. For Ag–Ag bonding, the optimum sintering temperature was 225 °C and a shear strength of 46.4 MPa was achieved. During the densification process of the hollow sintered structures, a large number of vacancies inside the sintered structure gradually diffused outward by increasing the concentration of vacancies in the grain boundaries, accompanied by a gradual decrease in the total atomic number of the grains. In addition, after the densification was completed, the diffusion pattern of grain boundary migration was activated and the size of the large grain gradually increased during the sintering process. However, the shear strength of the sintered structure was weakened due to the increase in the vacancy concentration of grain boundaries during the migration of the grain boundary. The results of our study enrich the Ag sintering mechanism.

## Figures and Tables

**Figure 1 micromachines-12-00521-f001:**
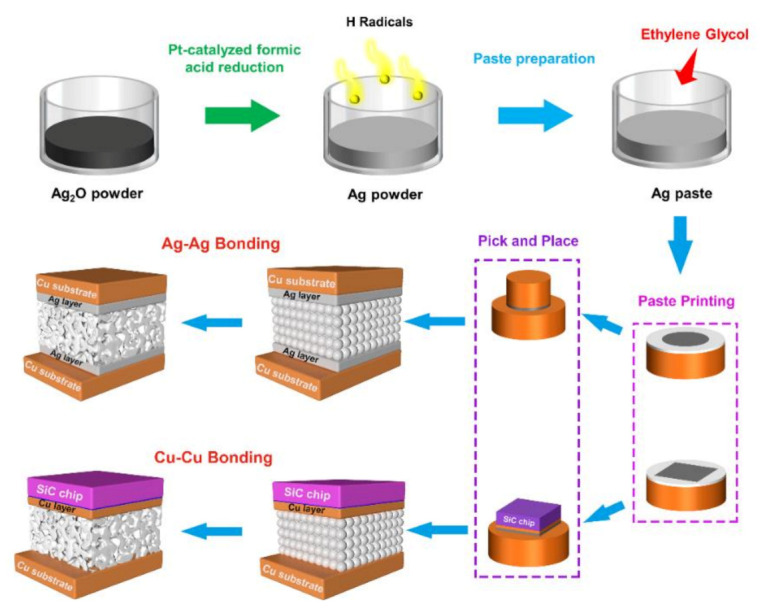
Schematic illustration of the experimental processes for Ag–Ag and Cu–Cu bonding.

**Figure 2 micromachines-12-00521-f002:**
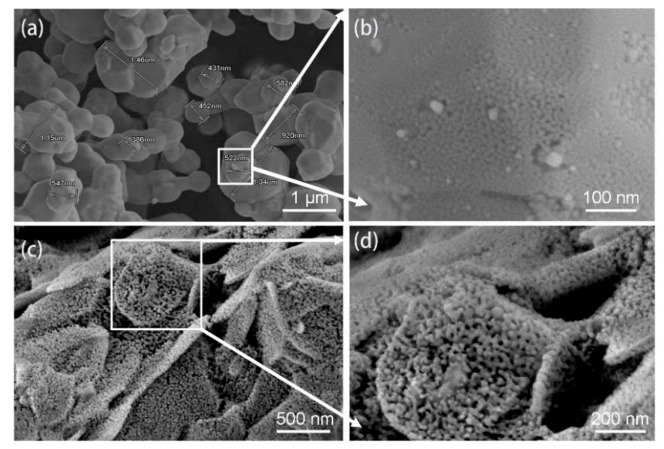
(**a**) SEM image of Ag_2_O powder, (**b**) high-magnification image of (**a**), (**c**) SEM image of Ag_2_O powder after the treatment in Pt-catalyzed formic acid vapor, and (**d**) high-magnification image of (**c**).

**Figure 3 micromachines-12-00521-f003:**
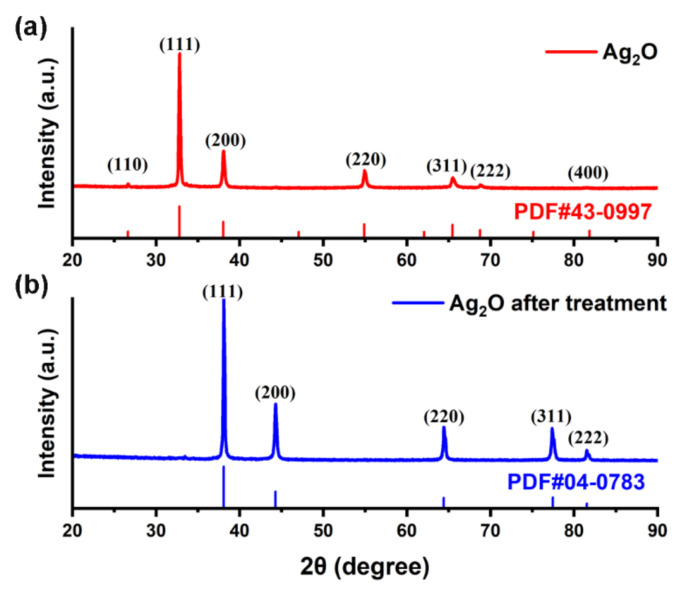
The XRD patterns of the following: (**a**) Ag_2_O powder and (**b**) Ag_2_O powder after the treatment in Pt-catalyzed formic acid vapor at 50 °C for 5 min.

**Figure 4 micromachines-12-00521-f004:**
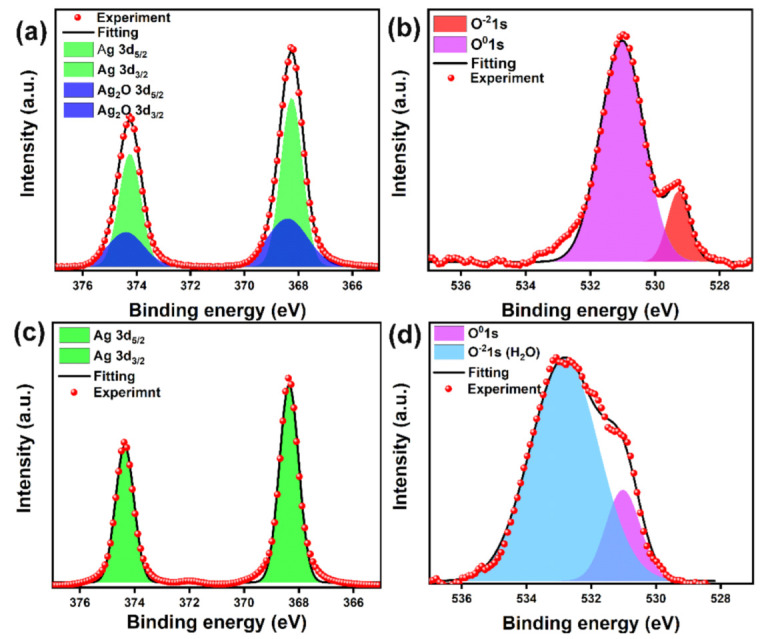
XPS analysis of Ag_2_O powder before and after a treatment in Pt-catalyzed formic acid vapor: (**a**) Ag3d spectra and (**b**) O1s spectra before the treatment, (**c**) Ag3d spectra and (**d**) O1s spectra after treatment.

**Figure 5 micromachines-12-00521-f005:**
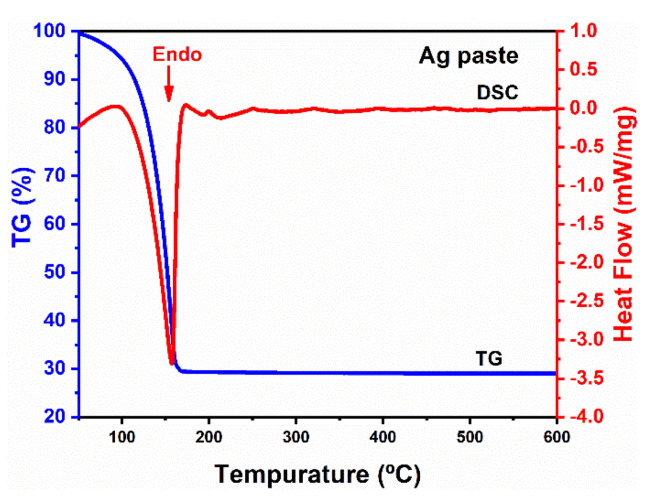
TG–DSC results of the Ag agglomerates paste heated in N_2_.

**Figure 6 micromachines-12-00521-f006:**
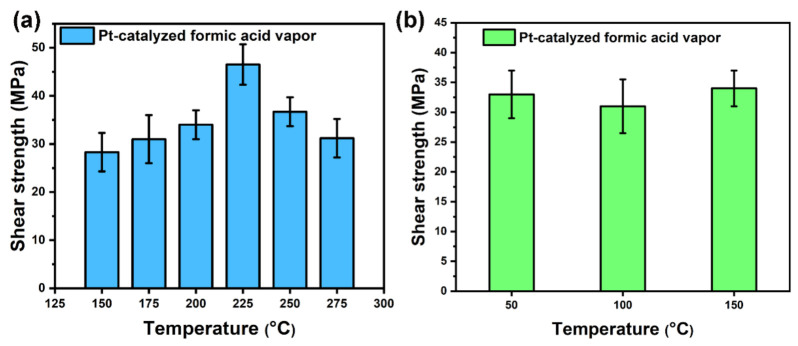
(**a**) The shear strength of Ag-plated copper substrates sintered at different temperature for 30 min in Pt-catalyzed formic acid vapor, and (**b**) the shear strength of Ag paste preheated at different temperatures for 5 min and then sintered at 200 °C for 30 min in Pt-catalyzed formic acid vapor.

**Figure 7 micromachines-12-00521-f007:**
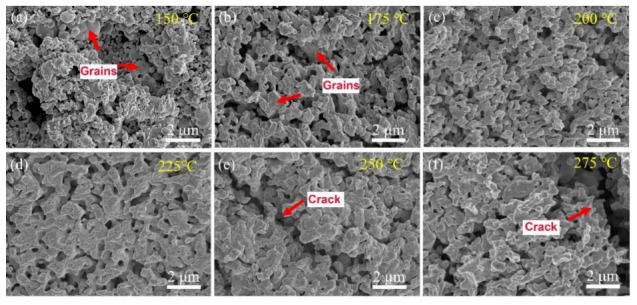
The SEM images of the fracture surface of specimens sintered at the following different temperatures: (**a**) 150 °C, (**b**) 175 °C, (**c**) 200 °C, (**d**) 225 °C, (**e**) 250 °C and (**f**) 275 °C.

**Figure 8 micromachines-12-00521-f008:**
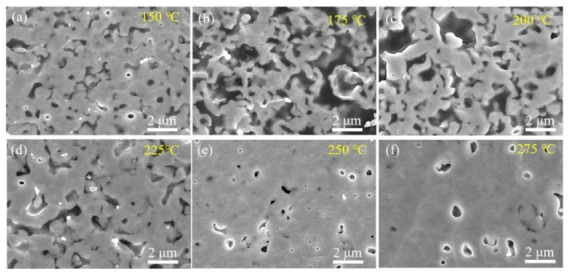
SEM images of cross-sectional structures of specimens sintered at the following different temperatures for 30 min: (**a**) 150 °C, (**b**) 175 °C, (**c**) 200 °C, (**d**) 225 °C, (**e**) 250 °C and (**f**) 275 °C.

**Figure 9 micromachines-12-00521-f009:**
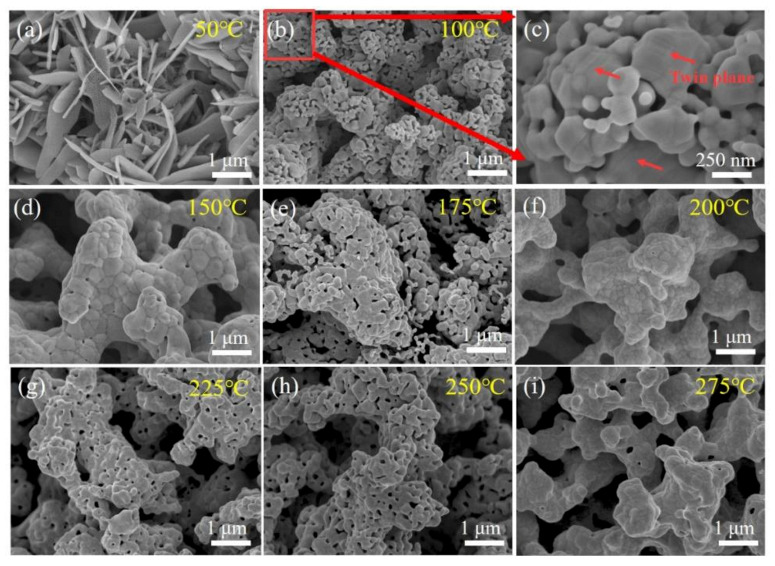
Sintering behavior of Ag agglomerates at the following different temperatures for 5 min under Pt-catalyzed formic acid vapor: (**a**) 50 °C, (**b**) 100 °C, (**c**) high-magnification image of (**b**), (**d**) 150 °C, (**e**) 175 °C, (**f**) 200 °C, (**g**) 225 °C, (**h**) 250 °C and (**i**) 275 °C.

**Figure 10 micromachines-12-00521-f010:**
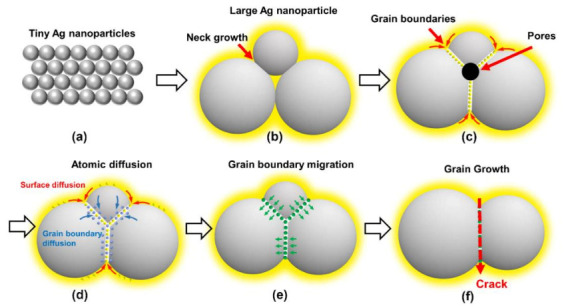
Schematic diagram of grain growth mechanism in surface of sintered structure. (**a**) the original morphology of Ag agglomerate, (**b**) formation of Ag nanoparticles with different sizes, (**c**) formation of pores at the position of triple junctions, (**d**) elimination of pores by surface diffusion and grain boundary diffusion, (**e**) movement of atoms from small size grain to large size grain by grain boundary migration, and (**f**) larger vacancies concentration and lower strength of grain boundary during further grain growth.

**Figure 11 micromachines-12-00521-f011:**
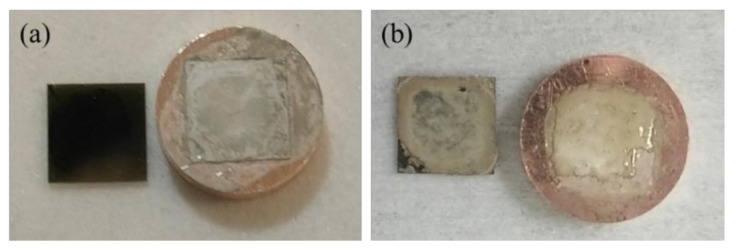
The photographs of different pastes after shear strength test. (**a**) Ag_2_O paste and (**b**) Ag agglomerates paste.

**Figure 12 micromachines-12-00521-f012:**
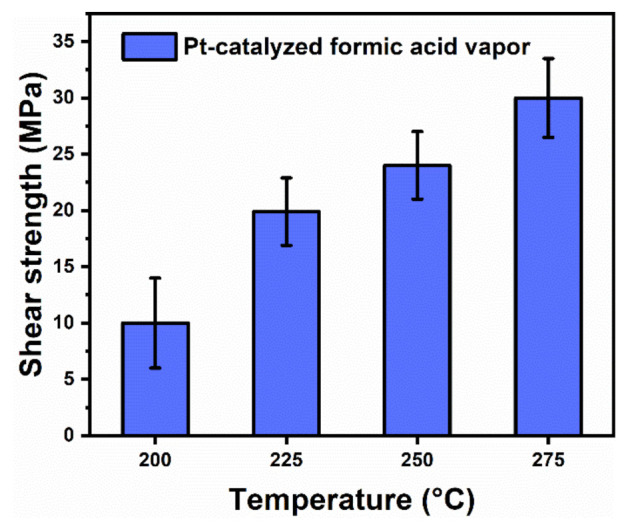
The shear strength of SiC/Cu–Cu substrates sintered at different temperatures for 30 min in Pt-catalyzed formic acid vapor.

**Figure 13 micromachines-12-00521-f013:**
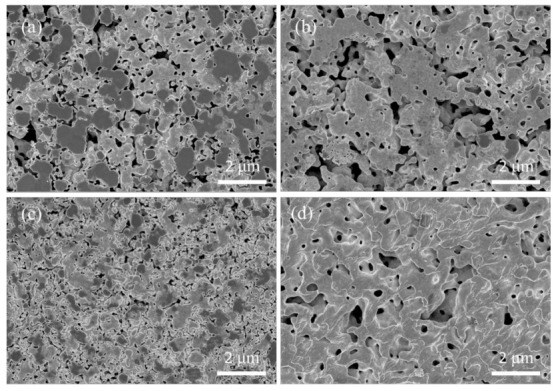
The fracture surfaces of Ag paste sintered at the following different temperatures: (**a**) 200 °C, (**b**) 225 °C, (**c**) 250 °C and (**d**) 275 °C.

**Figure 14 micromachines-12-00521-f014:**
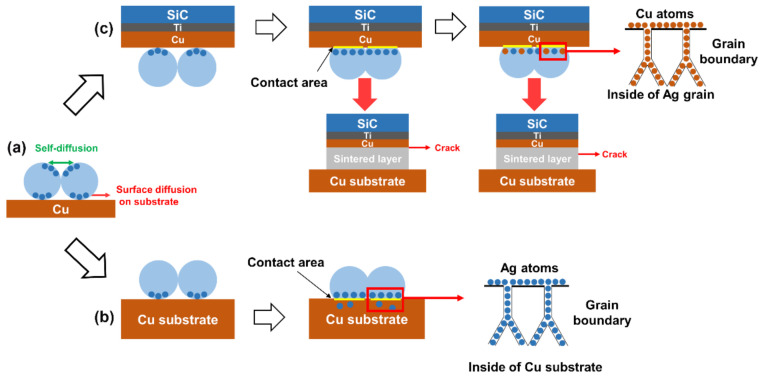
Schematic illustration of the diffusion in the Cu–Cu bonding. (**a**) Ag atoms diffusion during sintering process, (**b**) Ag atoms diffusion on the surface of bare Cu substrate, and (**c**) Ag atoms diffusion on the surface of Cu layer of SiC chip.

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
