# Peer review of "A Novel Preparation of Ag Agglomerates Paste with Unique Sintering Behavior at Low Temperature"

_micromachines, 2021, doi:10.3390/mi12050521_

Round 1

Reviewer 1 Report

The manucript, "A novel preparation of Ag agglomerates paste with unique sintering behavior at low temperature", describes the method to form Ag agglomerates paste from Ag2O by Pt-catalyzed formic acid vapor. The processed Ag paste can improve the bonding strength after sintering on Cu-Cu bonding. The manuscript can be published eventually after addressing my comments:

  1. There are a couple of format problems. Please read it carefully and correct it. For example, line 3 formatting needs to be updated. Line 10 "A" should not be bold. Line 262, "Tthe".
  2. It looks like line 65 Pt-catalyzed formic acid vapor is a critical step. Please add more information about how the vapor was purchased/formulated and how the process time and temperature will influence the bonding strength. 
  3. Please specify "special amount of EG" to mix the Ag particles and if the ratio will alter the behavior. What's the mixing process and instrument?
  4. What's the chemical reaction during the Pt-catalyzed formic acid vapor?
  5. Fig. 7(a), the failure mode is the adhesive between the Ag and SiC/Cu interface. Why? Does it happen at the Ag/Cu interface at the Cu substrate?
  6. Please add the chemical reaction equation after equ (1) for Fig. 11(b)
  7. What's the bonding pad surface finish for SiC chips in general? If it is Cu, why SiC chip didn't use other metals like Au or AlCu? Will this method show benefits for other type of metals?

Author Response

Dear reviewer,

Thank you for your comments concerning our manuscript entitled “A Novel Preparation of Ag Agglomerates Paste with Unique Sintering Behavior at Low Temperature”. Those comments are very helpful for improving our paper and inspiring our research. We have considered the comments very carefully and made modifications. The main corrections in the paper and the responses to the comments are as following:

Comments:

  1. There are a couple of format problems. Please read it carefully and correct it. For example, line 3 formatting needs to be updated. Line 10 "A" should not be bold. Line 262, "Tthe".
  2. It looks like line 65 Pt-catalyzed formic acid vapor is a critical step. Please add more information about how the vapor was purchased/formulated and how the process time and temperature will influence the bonding strength.
  3. Please specify "special amount of EG" to mix the Ag particles and if the ratio will alter the behavior. What's the mixing process and instrument?
  4. What's the chemical reaction during the Pt-catalyzed formic acid vapor?
  5. Fig. 7(a), the failure mode is the adhesive between the Ag and SiC/Cu interface. Why? Does it happen at the Ag/Cu interface at the Cu substrate?
  6. Please add the chemical reaction equation after equ (1) for Fig. 11(b)
  7. What's the bonding pad surface finish for SiC chips in general? If it is Cu, why SiC chip didn't use other metals like Au or AlCu? Will this method show benefits for other type of metals?

Responses:

  1. The format problems in the text have been corrected as following:
  1. The front size in line 6 has been changed into 8;
  2. The line 3 formatting has been updated: A Novel Preparation of Ag Agglomerates Paste with Unique Sintering Behavior at Low Temperature.
  3. Bold front of “A” in line 10 has been corrected;
  4. The writing mistake of “Tthe” in line 262 has been corrected;
  1. Relevant information has been added into the text: Anhydrous formic acid (99%) was purchased from Shanghai Macklin Biochemical Co., Ltd. And the temperature of Pt catalysis was 160℃ in our experiments. The entrance of formic acid vapor into the sintering chamber is via a flow of N2 and requires Pt catalysis prior to entering the chamber. In addition, the flow rate of the mixed gas was 40 cc/min.
  2. For the reduction of Ag2O powder, the reduction process should be conducted at lower temperature and shorter time conditions as much as possible. Otherwise, the tiny sized Ag nanoparticle would be slowly sintered together with the increase of temperature and time, resulting in the failure of the preparation of Ag agglomerates paste. In fact, the effect of temperature on the Ag agglomerate paste has been shown in Figure 9. Moreover, we believe that better results can be obtained by further reducing the temperature during the reduction process. However, since the catalytic unit and sintering chamber are connected, the temperature of the sintering chamber was around 46℃ under the working conditions of catalytic unit. Therefore, the temperature was set to 50℃ to avoid errors during the reduction process.
  3. The special amount of EG is 70 w.t%, which was explained in line 154. This data is based on the analysis of the TG-DSC results of the Ag agglomerates paste in Figure 5. Whether the ratio affects the sintering behavior was not investigated in the experiments. On the one hand, the proper ratio is conducive to the paste printing with steel mask. Too much solvent can not form a specific shape of paste, and too little solvent can cause the difficult in the control in the thickness of paste. On the other hand, the sintering chamber needs to be treated several times with evacuation equipment so that there is less solvent left. The paste was mixing Ag agglomerates powder with EG by a magnetic stirrer for 12 hrs.
  4. At 160℃, radicals are generated from formic acid by Pt catalysis,
  1. The results of Figure 7 were discussed about the sintering behavior of Ag-Ag bonding. The crack always appeared between the Ag sintered layer and SiC/Cu for all of the specimens, thus it related to the Cu layers of upper and lower substrates. Compared with the upper substrates’ Cu layers prepared by magnetron sputtering, bare copper substrates process a large number of grain boundaries. And these grain boundaries facilitated the migration of Ag atoms into the copper substrates and then enlarged the diffusion band (distance) together with the Ag atoms diffused from grain boundaries. Preceding study has shown metallic bonds are formed by inter-diffusion band of Cu-Ag. Therefore, the diffusion band contributed to the stronger metallurgical bonding on copper substrate than SiC chip. In addition, the surface of the bare copper substrate is rougher and the contact area is larger compared with the copper layer of SiC chip.
  2. The chemical reaction equation for Fig. 11b has been added:
  3. The operating temperature of power devices is relatively high (above 275℃), and it is necessary to select the metal material with high electrical conductivity and thermal conductivity. The metal layer finished for SiC chip are Au, Ag and Cu in general, and their thermal conductivity are 315 W/m·K, 419 W/m·K and 401 W/m·K, respectively. However, it was found that the diffusion rate of Ag atoms into the Au metal layer is high, resulting in the formation of cracks at the interface which the reduced reliability of the power devices in the process of practical application[1]. Therefore, the Ag and Cu finished layer were investigated in our experiments. However, whether this method is beneficial for other type metals has not been studied.

Reference:

  1. Wang, X.; Mei, Y.; Li, X.; Wang, M.; Cui, Z.; Lu, G., Pressureless sintering of nanosilver paste as die attachment on substrates with ENIG finish for semiconductor applications[J]. Journal of Alloys and Compounds 2018, 777, 578-585. 
  2. We have earnestly revised the manuscript accordingly. The revised portion is marked in highlight. Thanks again for your valuable comments and suggestions.
  3.  

Sincerely yours

                                                       Junlong Li and Yang Xu

                                                     E-mail: xuyang@ime.ac.cn

Reviewer 2 Report

In this study, the authors investigated a Ag agglomerates paste for Cu-Cu bonding and Ag-Ag bonding, and they found out the optimum condition at 225 Celsius degrees for Au-Au bonding, which resulted in a shear strength of 46.4 MPa, and Cu-Cu bonding resulted in 20 MPa. They also discussed the sintering mechanism of the bonding formation, focusing on Ag agglomerates. I agree that this study attracts attention of readers of Micromachines. However, I would like to ask the authors to improve the manuscript before acceptance for publication.   

  1. Figure captions should be described in more detail. For example, the authors should describe explanations of each panel in Figures 3, 7, 8, 9, 10, 11, and 14.

  1. In Figure 2, I guess panels (b) and (d) are magnified view of (a) and (c), respectively, but the line segments drawn from the panel (a) to (b) and those from the panel (c) to (d) seem to be poor, which may be misunderstood. You should improve them and clearly describe the correspondence at the figure caption. Dashed line segments in Figure 9(b) is similarly unclear.

  1. How many samples did you measure to evaluate averages and error bars in Figures 6 and 12.

  1. At Line 246, although the authors mention Figure 10g, 10h, and 10i, those figures are not in the manuscript.

--

Author Response

Dear reviewer,

Thank you for your comments concerning our manuscript entitled “A Novel Preparation of Ag Agglomerates Paste with Unique Sintering Behavior at Low Temperature”. Those comments are very helpful for improving our paper and inspiring our research. We have considered the comments very carefully and made modifications. The main corrections in the paper and the responses to the comments are as following:

Comments:

  1. Figure captions should be described in more detail. For example, the authors should describe explanations of each panel in Figures 3, 7, 8, 9, 10, 11, and 14.
  2. In Figure 2, I guess panels (b) and (d) are magnified view of (a) and (c), respectively, but the line segments drawn from the panel (a) to (b) and those from the panel (c) to (d) seem to be poor, which may be misunderstood. You should improve them and clearly describe the correspondence at the figure caption. Dashed line segments in Figure 9(b) is similarly unclear.
  3. How many samples did you measure to evaluate averages and error bars in Figures 6 and 12.
  4. At Line 246, although the authors mention Figure 10g, 10h, and 10i, those figures are not in the manuscript.

Responses:

  1. Figure captions have been modified as shown following:Figure 7. The SEM images of fracture surface of specimens sintered at different temperatures: (a) 150℃, (b) 175℃, (c) 200℃, (d) 225℃, (e) 250℃ and (f) 275℃.Figure 9. Sintering behavior of Ag agglomerates at different temperature for 5 min under Pt-catalyzed formic acid vapor: (a) 50℃, (b) 100℃, (c) high magnification image of (b), (d) 150℃, (e) 175℃, (f) 200℃, (g) 225℃, (h) 250℃ and (i) 275℃.Figure 11. The photographs of different paste after shear strength test: (a) Ag2O paste and (b) Ag agglomerates paste.
  2. Figure 14. Schematic illustration of the diffusion in the Cu-Cu bonding: (a) Ag atoms diffusion during sintering process, (b) Ag atoms diffusion on the surface of bare Cu substrate and (c) Ag atoms diffusion on the surface of Cu layer of SiC chip.
  3. Figure 10. Schematic diagram of grain growth mechanism in surface of sintered structure: (a) the original morphology of Ag agglomerate, (b) formation of Ag nanoparticles with different size, (c) formation of pores at the position of triple junctions, (d) elimination of pores by surface diffusion and grain boundary diffusion, (e) movement of atoms from small size grain to large size grain by grain boundary migration and (f) larger vacancies concentration and lower strength of grain boundary during further grain growth.
  4. Figure 8. SEM images of cross-sectional structures of specimens sintered at different temperatures for 30 min: (a) 150℃, (b) 175℃, (c) 200℃, (d) 225℃, (e) 250℃ and (f) 275℃.
  5. Figure 3. The XRD patterns: (a) Ag2O powder and (b) Ag2O powder after the treatment in Pt-catalyzed formic acid vapor at 50℃ for 5 min.
  6. Figure 2 and Figure 9 were modified as following:Figure 9. Sintering behavior of Ag agglomerates at different temperature for 5 min under Pt-catalyzed formic acid vapor: (a) 50℃, (b) 100℃, (c) high magnification image of (b), (d)150℃, (e) 175℃, (f) 200℃, (g) 225℃, (h) 250℃ and (i) 275℃.
  7. Figure 2. (a) SEM image of Ag2O powder, (b) high magnification image of (a), (c) SEM image of Ag2O powder after the treatment in Pt-catalyzed formic acid vapor, and (d) high magnification image of (c).
  8. For every condition, 5 specimens were measured for Figure 6 and Figure 12.
  9. “Figure10g, 10h and 10i” has been modified to “Figure 9g, 9h, and 9i” in the line 246.
  10.  

We have earnestly revised the manuscript accordingly. The revised portion is marked in highlight. Thanks again for your valuable comments and suggestions.

Sincerely yours

                                                      Junlong Li and Yang Xu

                                                     E-mail: xuyang@ime.ac.cn
